# Harm Received, Harm Caused: A Scottish Gael's Journey to Becoming Pākehā

Dani Pickering

School of Social and Cultural Studies, Victoria University of Wellington, Wellington 6012, New Zealand; dani.pickering@vuw.ac.nz

**Abstract:** *Beurla an donais*. The language of the devil. This is how my great-great-great grandfather, Neil McLeod, described English in his native Gaelic as he grieved the loss of his wife Rebecca Henry in 1886. Even as he tried to distance himself socially and linguistically from the Anglophone world, however, he had already long since been caught up in its colonial machinery. After being cleared from his ancestral homeland of Raasay, Scotland in 1864 and relocated to the colonial frontier in Aotearoa New Zealand, Neil went on to spend more than fifteen years in the New Zealand Armed Constabulary and its reconstituted form, the New Zealand Police Force, before being killed on the job in 1890. Drawing on critical family history literature, firsthand accounts from Neil's personal diaries, other family accounts and additional historical research, this article retraces Neil's assimilation into white New Zealand. By unsettling the "constitutive forgetting" by which Neil and his descendants forsook our connection to Raasay and the Scottish *Gàidhealtachd* to become Pākehā settlers, I explore a history prior to and concurrent with the colonisation of Aotearoa which accounts for multi-ethnic Pākehā origins, beyond the Anglo-Saxon, and enables a deeper understanding of how and why Gaels such as Neil participated in the British Empire. I conclude by considering how Neil's story deepens our understanding of how the settler-colonial subject is produced by highlighting the occasionally fine but always distinct line between coloniser and colonised.

**Keywords:** assimilation; British Empire; colonisation; settler-colonialism; critical family history; Gàidhlig; Gaelic; Pākehā



## 1. Introduction

Critical family history research is a powerful tool for distinguishing between larger historical narratives and the smaller stories of the individuals participating in that history. While the two are invariably linked (Buchanan 2012), there are nevertheless tensions between these two approaches to the past; when is it necessary to connect family histories to the readily apparent outcomes of "big" stories such as colonisation (Shaw 2021a), and when is it "unfair to deduce an intent from the sweeping, systemic forces of history and impute it to a particular individual" (Shaw 2022)? When "big" history is not just the "backdrop on which to locate [one's] ancestors" but incorporated as "part of the family story" (Sleeter 2020, p. 1), a more nuanced response to these kinds of questions becomes possible.

This nuance is particularly helpful for making sense of situations such as those in Aotearoa, where the Pākehā (white settler) population has imposed the settler-colonial nation-state of "New Zealand" on Māori society since the mid-nineteenth century (Anderson et al. 2015; Awatere 1984; Belich 2002; Walker 2004). These original Pākehā settlers came from a range of not only national but ethnic backgrounds—English, Welsh, Lowland Scots, Highland Gaels, Irish Gaels, and Ulster Scots—some of which had previously been colonised by and subsumed into the British Empire (Lenihan 2015; MacKenzie 2008; MacKinnon 2008; MacKinnon 2017; McCarthy 2017; Newton 2015).

Though long since absorbed into a monolithic white settler identity (Bell 1996; Pearson 1989), this ethnic diversity within the original Pākehā population no doubt lent itself to a variety of motivations, intentions, and rationales for participating in British Imperial expansion in Aotearoa. For Celtic-origin Pākehā, however, contributing to and benefitting from the dispossession of Māori would appear to be particularly hypocritical, as ostensibly colonised peoples going on to become colonisers. This apparent heel-turn begs what I call the "Celtic question": if Pākehā who migrated from nations such as Wales, Ireland, and Scotland did indeed come from colonised backgrounds, what prevented them from seeing common cause with Māori seeking to protect their own sovereignty from the British Empire? Put more simply, how could the dispossessed go on to dispossess others?

In this article, I explore this question by using a critical family history framework to examine almost two decades' worth of diaries written by my great-great-great grandfather Neil McLeod. Written bilingually in English as well as his "mother tongue" *Gàidhlig* (Scottish Gaelic, hereafter referred to as Gaelic), the diaries cover the majority of his time as a first-generation Pākehā settler in Aotearoa, from his initiation into the New Zealand Armed Constabulary (AC) in 1871 to a few years before his death in 1890. Throughout, Neil grapples with his identity as *Gàidheal* (a Scottish Gael) while he slowly but surely shows both implicit and explicit signs of his assimilation into the Anglo-Pākehā world.

The article begins with a review of the critical family history literature that informed the project framework and methods, as well as introducing the research materials (i.e., Neil's diaries). Then, I provide a brief and highly condensed overview of Scottish Gaelic history up to and coinciding with the initiation of colonisation in Aotearoa, so as to introduce Neil's ethnic background and its significance to the study. Then, I analyse the diaries themselves, covering broad findings while zooming in on passages that are particularly revealing of what Neil thought or ignored about his role in the settler-colony, its impact on his identity, and his impact on others. The article concludes with a discussion of the implications of this research for a multi-ethnic understanding of British colonialism in Aotearoa, directly addressing the "Celtic question" of Pākehā identity posed above.

## 2. Methods and Materials

Bruce Durie distinguishes family history from genealogy by understanding the latter as "the basic data on which further socio-historical and narrative information is built" (Durie 2017, p. 2), ranging from names, dates and places of births, marriages, and deaths to other contextual details such as occupations and migration patterns. Family history, then, takes these data and constructs a narrative from them by drawing on the social, cultural, political, economic, and other contexts of the relevant period of history to "describe or explain what the family did within specific contexts, and how that context mattered" (Sleeter 2020, p. 2).

Family history is, also, by its very nature "the first history many of us learn", and as such has a profound capacity to shape our understandings of the world (Shaw 2021a, p. 2). As a result, it says just as much with what it *excludes* as what it includes, and as critical theory teaches us, this makes family history inherently political (Sleeter 2020, p. 2). The application of critical theory to these histories is especially relevant in settler-colonial contexts such as Aotearoa New Zealand. Here, uncomfortable truths about our Pākehā ancestors' culpability in the colonisation of the Māori world are typically conveniently omitted through a "constitutive forgetting" of our true homelands and how we came to be in Aotearoa, consolidating a white settler Pākehā identity which reinforces the colonial status quo (Bell 2020, p. 3; Borell et al. 2018, p. 28; Connerton 2008).

*Critical* family history therefore reintroduces many of these omitted elements. In addition to acknowledging the personal impact of "big" stories such as colonisation, critical family history introduces concepts such as "footholds," or economic opportunities more privileged ancestors acquire to increase their personal wealth and further consolidate that privilege, as well as "cushions", through which a family's inheritance of previous generations' gains can shield that family from the impact of major life setbacks (Sleeter 2014).

Together and across generations, these footholds and cushions accumulate as "historical privilege", which in the Aotearoa New Zealand context is fuelled by the continued dispossession and exploitation of Māori (Borell et al. 2018). As a result of this dispossession and exploitation, Māori consequently experience the more widely understood "historical trauma", which is distinctive for being collectively experienced and multigenerational (p. 25). By acknowledging these dynamics, critical family history research has the capacity to bridge the "little" and "big" stories of history by demonstrating how privilege functions over the long-term, and at whose expense.

Avril Bell concedes, however, that research into these footholds and cushions tends to itself privilege analysis of "capitalist/material forms of value", rather than engaging more comprehensively with other value systems and worldviews (2020, p. 2). This emphasis is not without good reason; material wealth (or lack thereof) invariably plays a major role in shaping lives (Borell et al. 2018, p. 28), as does control of the lands from which that wealth is produced (Coulthard 2014; Poata-Smith 2002). This emphasis on a more material analysis in critical family history research is also informed by the available *research* materials. Most of those used to reveal various footholds and cushions are a matter of public record (land titles, wills, purchase records, etc.) (Sleeter 2014; Bell 2020; Shaw 2021a; Shaw 2021b). However, what analysis of these materials typically cannot do is reflect the perspectives and agencies of the ancestors at the centre of such research. As Richard Shaw emphasises, it is difficult in research on the "little" stories of those ancestors to do anything more than project perspectives and intentions onto them (Shaw 2021a, p. 8; 2022); their lives were as diverse as our own, and they were therefore no less likely to digress from the "big" historical narratives than we are. While public records do clearly signal intent (intent to buy, to sell, to inherit, etc.), they are far less likely to demonstrate the rationale, motivations, or even emotions behind that intent which distinguish the "little" stories of family histories. It is through private records such as personal diaries, then, and the stories they tell, that a more diverse range of expressions is likely to emerge (Alaszewski 2006).

Neil's two diaries are an exemplary case study for these more private perspectives. The first comprises approximately[1] 500 entries from the day he first joined the AC on 1 July 1871, up until the day of his wedding with Rebecca (née Henry) on 29 May 1878. The second diary has fewer entries (at 80), but they are longer and look back over more than one day at a time. Through this more retrospective approach, his narrative picks up right after his and Rebecca's wedding, focusing on his life with her until her death in 1886, his remarriage to Elizabeth (née Parkinson) a year later, and to their family's move to Dargaville, where the second diary ends mid-sentence in an entry dated 14 March 1887.

Not only are these diaries a record of Neil's personal thoughts over an extended period of time, but their bilinguality provides additional layers of expression which enable a more complex analysis. In other words, what he chooses to write about as well as *which language* he chooses to write about it in reveals much about his position within his social and emotional worlds. There are roughly 60 entries across both diaries which feature Gaelic and therefore enable this analysis, and a translation of most of them is included with the diary copies held at the Auckland War Memorial Museum (McLeod 1871–1887). However, as Michael Newton argues, understanding Gaelic perspectives on their own linguistic terms is essential to any engagement with the Gaelic world, especially given (a) how much of Gaeldom's cultural and intellectual resources have been poured into literary pursuits over the centuries and (b) how thoroughly this literary corpus "has been ignored in, or distorted by, the official national narratives" (2015, p. 9). While I was therefore able to draw from the translations available, I nevertheless resolved to engage with the Gaelic passages directly by drawing from my own years of study of the language. Attempting my own translations, several of which are included in this article, ensured that I was better able to understand and appreciate nuances to Neil's expression which would have been obscured without engagement with and through the Gaelic language itself.

To analyse these diaries with regard to the "Celtic question", I treated them as representations of an "external reality" in order to "develop an understanding or theory about

this external reality" (Alaszewski 2006, p. 51). Through this approach, I was less interested in confirming the factual accuracy of everything Neil described; rather, I sought to reveal more of Neil's thoughts and feelings about his social world through *how* he wrote about it. While this focus on the diaries as a "creatively constructed narrative" (p. 44) did enable me to sidestep questions of historical accuracy to some extent, I still had a responsibility to consider what Neil excluded from that narrative as much as what he included (Schoppa 2010, pp. 31–32). For example, I measured the decline of Neil's use of Gaelic between the first and second diaries, carefully considering what that revealed about his assimilation, and also noted how his descriptions of work in the AC frame or omit acts of violence that are known to have been part of that work (Hill 2000).

To the critical family history researcher, these "little" stories have much to say about the larger historical narratives within which Neil is positioned; conversely, so too are Neil's writings during his time in Aotearoa better understood through the introduction of additional social, cultural, political, economic, and, even, linguistic contexts which shaped his life before and after his migration. The following section will therefore establish some of those contexts through a brief overview of Scottish Gaelic history, before using it to present a critical family history.

## 3. Whence Neil Came

Neil was born 15 June 1846 on Raasay, a small island in the Scottish Hebrides. The Hebrides are part of a region of Scotland called the *Gàidhealtachd*, which refers not to a geographic boundary so much as to a linguistic one within which Gaelic is the primary language spoken (MacInnes 2006, p. 104; Newton 2019, pp. 51–52); in the Anglophone world, this region is more commonly referred to as the Scottish Highlands and Islands. Though the Gaels of the Irish kingdom of Dàl Riata are associated with the founding the Kingdom of Scotland itself in the face of competition from the Britons, Picts, and Angles, this primacy would not last; once Inglis (the Early Scots language) replaced Gaelic as the lingua fracas of the Scottish courts in the late eleventh century AD, Scottish Gaelic language and culture began a long, slow decline (MacInnes 2006, pp. 92–96; Newton 2019, p. 15–21).

This decline accelerated in the eighteenth century as a result the Jacobite Risings, a series of conflicts initiated by the ousted Stuart family who sought to reclaim the English throne from the Hanoverian monarchy (Newton 2019, pp. 33–34). Many Highland clans, including Clan MacLeod of Raasay, backed the Stuart claim to the throne (MacLeod 2002, pp. 51–52), though they were decisively defeated at the Battle of Culloden in April 1746. Culloden marked a turning point for the worse for the Gaelic world; partly as retaliation for the clans' participation in the Risings, and partly in accordance with its emerging capitalistic, expansionist values, Britain began efforts to "civilise" the *Gàidhealtachd* and claim its assets for the Empire (MacKinnon 2017, p. 38; Newton 2015, p. 33; Newton 2019, p. 36). This "civilising" effort was seen as necessary because while the Scottish Highlands were territorially considered part of the United Kingdom by this point, Gaels from the Highlands were not only considered (and considered themselves) a separate ethnic group within Scotland but were explicitly racialised as savages by Anglo-Saxons, including Lowland Scots (MacKinnon 2017, pp. 34–35; MacKinnon 2019; Stroh 2016, pp. 185–211). Consequently, efforts to "civilise" the *Gàidhealtachd* were explicitly treated by the Empire as "domestic colonization", the components of which would go on to inform the blueprint for its colonial projects globally from the late eighteenth century onward (MacKinnon 2017, p. 32).

In the *Gàidhealtachd*, this "domestic colonization" took the form of the Highland Clearances[2], a century-long process by which everyday Gaels were (often forcibly) evicted from their ancestral lands in favour of more economically productive uses of those lands (e.g., sheep grazing for the then-booming wool economy) (MacKinnon 2008, p. 6; MacKinnon 2017; Newton 2015, pp. 34–36; Newton 2019, pp. 37–38). Previously, land in the *Gàidhealtachd* was regarded in terms of its "social potential—the number of people it could support" (Newton 2019, p. 37). In order to integrate into the mainstream capitalist economy,

however, landholders in the Highlands had to grow their enterprises by consolidating land into larger and larger individual holdings, ultimately prioritising profit over social potential (Grant 2016, p. 23; Newton 2019, p. 37; Stroh 2016, p. 100). Consequently, everyday Gaels did not profit from such ventures; rather, the architects of the Clearances, Lowland entrepreneurs and the landholding élite of the Highlands, the clan chiefs, did. With the passing of the Statutes of Iona in Scotland in 1609, clan chiefs had already become obligated to send their heirs to be educated in the Lowlands, where they were "anglicised" through an English-language education which taught English (and thus capitalist) rather than Gaelic values (Newton 2015, p. 33; Newton 2019, p. 31; MacInnes 2006, pp. 106–7; Meighan 2020, p. 2; Stroh 2016, p. 75). Coupled with the increasing commercial pressures of a modernising Scotland, the divide-and-rule tactics of anglicisation severed the social bonds between chief and clan, reframing social classes in Gaelic society through a capitalist lens and thereby producing the rationale for clan chiefs to evict their own people (Newton 2015, p. 36).

The "civilising" of everyday Gaels was more often conducted through military service, which was achieved in part through the reframing of Gaels as noble savages from a supposedly "warrior" culture who were therefore particularly well-suited to serving the British Empire as soldiers (McLeod 2013; Newton 2008; Newton 2015, pp. 68–78; Newton 2019, pp. 39–40; Stroh 2016, pp. 113–40). This reconstruction of Gaelic identity itself was "carefully crafted and nurtured by the 18th-century land-owning elite for their own benefit", since landlords could then "exploit tenants for military recruitment" (Newton 2015, p. 69; Newton 2019, p. 38). The result was a sociopolitical context in which everyday Gaels were effectively presented with two choices: either participate in the expansion of the British Empire by emigrating to its more distant settler-colonies (and fighting in its wars there), or choke on the fumes of the then-industrialising urban centres of the Scottish Lowlands.

In the century after Culloden, this choice was presented to the *Gàidhealtachd* writ large as the Clearances unfolded. Raasay was one of the last places affected, however, with major emigration waves not taking place until the 1850s and 1860s when my great-great-great-grandfather was coming of age there (MacLeod 2002, pp. 104–17). As Neil and his family left the island for the last time in 1864, his father Murdoch reportedly "took his bonnet from his head, hid his face in it and wept" as they said their final goodbyes to their homeland (Smyth 1965, p. 7). A few months later, Murdoch and three of his five brothers and their families arrived in Aotearoa aboard the *Viscount Canning* on 21 January 1865, just over a century after the Highland Clearances began the colonisation of the *Gàidhealtachd* in earnest.

## 4. The Life and Times of Neil McLeod

Education on Raasay within Neil's lifetime was sponsored by the Scottish Society for the Propagation of Christian Knowledge (SSPCK), an anglicising mission which deployed schoolmasters to townships across the *Gàidhealtachd* to teach communities "the elements of reading and writing" (MacInnes 2006; MacLeod 2002, pp. 83–84). Though one of its purposes was to anglicise, and therefore to marginalise the Gaelic language, the SSPCK ironically played a role in establishing a more *Gaelic*-literate *Gàidhealtachd*, since it had to translate Biblical Scriptures into Gaelic to successfully engage Gaels with its programmes (MacInnes 2006, p. 439). Moreover, even as Neil would have been taught to read and write *English* through SSPCK schooling, teaching conditions on Raasay at the time indicate that actual instruction was most likely almost exclusively in *Gaelic* (MacLeod 2002, p. 83), necessitating some degree of literacy in both languages.

About six and a half years after his arrival in Aotearoa, Neil makes use of this literacy by beginning to document his life in his diaries. From 1 July 1871, he covers events from the "little" life events that changed *his* world, to the mundane and everyday activities that nevertheless played a role in reshaping society, to the "big" stories that were shaping the wider world at the time. Each of these three categories is in its own way revealing of Neil's evolving attitudes toward himself, his people, other peoples, and his role in the Empire.

*4.1. Na Sgeulachdan "Beaga"/the "Little" Stories: A Gael in An Anglo-Pākehā World*

The majority of Neil's diary entries are written in English. However, given Neil's likely bilingual schooling, his predominately English writing is more likely a reflection of the fact that he is now in a situation where English, rather than Gaelic, is the language crucial to "forming and nurturing the bonds of family and community" in his everyday life (Newton 2015, p. 377). The significance of this change should not be understated—language is tightly interwoven with sense of self and the surrounding environment which produces that self (Nash 2018, p. 359; Stewart 2020, p. 23). Indeed, *Gàidhlig* and the *Gàidhealtachd* are such integral parts of Gaelic identity that before the nineteenth century, *Gàidheal* simultaneously referred to both Gaelic speakers and inhabitants of the *Gàidhealtachd*; there was no distinction (Newton 2015, p. 6; Newton 2019, pp. 51–52; see also Bechhofer and McCrone 2014). Therefore, Neil takes his most foundational steps toward becoming Pākehā simply by leaving the *Gàidhealtachd* and using English more than Gaelic in order to participate in his new community.

However, he does not abandon his mother tongue wholesale just because he is no longer surrounded by other Gaelic speakers. Rather, he finds new purposes for the language in his colonial life which are unique to the diasporic context:

[18th September 1877] Tha eagal orm gum beil am fear seo a faicinn ciod a tha mi sgriobhadh, ach airson a tha cinnteach sgriobh mi e ann a ghailic.

[I am afraid that this man is seeing what I am writing, but to be sure I am writing it in Gaelic.]

Thanks to the above passage, we can reasonably infer that which language Neil writes in is not only preferential but strategic: he takes advantage of Gaelic's minority status in Aotearoa to afford his written thoughts greater privacy when he deems it necessary. Consequently, the rest of his Gaelic writing features less mundane commentary on the weather and other daily happenings such as the one above, and more his more private thoughts about people, his feelings toward them, and his relationship to faith, God, and his own identity.

For example, Neil records his thoughts and dreams about, and waking experiences with women almost exclusively in Gaelic (in the first diary at least). Many of his more romantic and even lustful preoccupations for several years centre on a woman named Beasa:

[7th May 1876] Chunnaic mi aisling agus is e seo i, Bha mi a coiseachd ann an aite [ainethach?]. chunnaic mi ban-charaid agus Beasa maille rithe, bha Beasa leth lormachd, agus thuit i ann an tobar-domhain. chuidich mise i as an tobar agus thoisich i ri labhairt-Gaelic rium! Cha' n eil chuimhne agam ciod e a bha i a labhairt no cea i an t-aite anns an robh sinn.

[I had a dream and this is it. I was walking in a [recognisable?] place, I saw a ladyfriend and Beasa alongside her, Beasa was half naked, and she fell in a deep well. I helped her from the well and she started speaking Gaelic to me! I cannot remember what she was saying nor what place we were in.]

[13 May 1876] Chunnaic mi bruadar o' chionn oidhche no dha, air am [x] seo "Bha Beasa agus Anna a piutheir seo n' suidhe air bruach loch. Shuigh mise ri taobh Beasa agus thoisich mi ri cleachd rithe, thuit i air a druim agus thuit cuid dhi a h-aodach dhi, leam i an-sin anns an loch agus thoisich ise air snàmh"

[I saw a vision a night or two ago, on this [x]: "Beasa and Anna her sister were seated on the edge of a lake. I sat next to Beasa and started to make fun of her, she fell on her back and part of her clothes fell off her, she then jumped in the lake and had started swimming.]

While the intimacy of the content of these dreams is no doubt one reason he writes about them in the greater privacy of Gaelic, what they also reveal in the full context of the diaries is that despite losing Gaelic as a community language, it has found continued purpose as a

language of and *for* intimacy. In this way, Neil finds a refuge for his mother tongue which he largely manages to preserve throughout the 1870s.

Beyond this newfound purpose for the language, the deferential way Neil regards Gaelic and other Scottish Gaels, even when writing about them in English, shows how important aspects of Gaelic identity remain to him even into his later years:

> [20 January 1886] I could not hold my thoughts and found some relief in the wildest exclamations, but I could not find English expressive enough. I tried my mother tongue, Gaelic, suited me, its sympathising words and expressiveness in some measure relieved the awful load with which I was burdened, but in my ravings I went so far as to say "Beurla an donais" [English/language of the devil][3] more than once.

> [20 January 1886] Mr Thomson's appearance at daylight somewhat relieved me in a few Gaelic words, he was kind enough to take two telegrams for me to be sent to Rev Warren and father. Father's was in Gaelic.

> [21 January 1886] On the arrival of the steamer, father, Rory, Uncles Joseph & Young and Frank arrived and consoled me as well as they could. My poor friend John Bannatyne was very attentive to me and proved indeed to be a real and true hearted Highlander.

These entries from the second diary narrate the aftermath of the death of Neil's first wife and my great-great-great grandmother Rebecca. In them, it is clear how Neil's grieving has reached the point at which "the significance of the homeland identity became fundamental" (Powell 2007, p. 92). While other entries throughout the first diary do indeed demonstrate Neil's preference to be in the company of women and/or other Scottish Gaels, these passages most explicitly highlight how important the Gaelic language and people are to him, as they are who and what he draws the most solace from in more trying times.

In contrast, Neil tends to regard the people who are neither Gaels nor women he fancies quite critically, perhaps disproportionately so given the additional privacy Gaelic affords him in the diasporic context. While on occasion the reasons for these criticisms appear obvious (an entry on 10 August 1876 criticises a cook for serving raw and/or rotten meat), in most other cases the reasons are more values-based and therefore vague:

> [10 July 1876] Tha mi fas gle sgith dhe 'a uile seo, oir tha a chuid is seo don t-sluagh gle mi-bheusach agus buaireasach.

> [I am growing tired of all of this, because most of these people are very unvirtuous and troublesome.]

> [5th August 1876] Tha an Donas air a choisibh ann san aite-seo, cinnteach gu leir. tha cuid do mhuinntir a bhaile cho buarrasach + cho buaireasach agus cho mi-bheusach, 's nach urrain duine a bheul fhaghadh, tha do bhreugan mun dhiu fein agus san chairdain air fegh a bhaile is [x] duthaich.

> [The devil has walked in this place, surely enough. Some people of the town are so troublesome and so unvirtuous, that a person cannot open his mouth among them, for the lies are about themselves and my friends in the town and the country.]

It is worth noting that most Pākehā at this time were of Protestant faiths as much as Neil (Stenhouse 2018). Having proudly identified himself as a Presbyterian in the very first entry of the first diary, Neil consistently evokes distinctly Christian language when making value judgments such as the ones above, which suggests at least some degree of common ground. However, because Presbyterianism is a distinctly Scottish denomination of Protestantism, and given Neil's previously demonstrated affinity for Gaels as a people, it is very possible that the disdain for other Pākehā he expresses here may be not only the product of a perceived lack of moral piety, but potentially informed by national and/or ethnic dimensions as well.

These distinctions are only ever implied, however. As Shaw points out, the "rhetoric of 'settlement'" does a lot of work for the colonial project by "collapsing old enmities... into a new cleavage between the white/coloniser/civiliser and the non-white/Māori/savage" (Shaw 2021a, p. 7). While Neil continually expresses affinity for his Gaelic brethren and disdain for most others throughout both diaries, he also quite startlingly recognises that living on the colonial frontier in Waikato, surrounded primarily by Anglo-Pākehā, is hastening his assimilation into their world:

> [26th July 1876] Tha mi fas gle sgith don aite-seo, a' measg sluaigh aingidh air nach eil eagal Dhe na dheomhain, no curaim air siorraidheachd no maith an annam no thuig Bruidean an t-shubha?, agus tha mi duilich innsidh gun bhith mi fin a fas cosmhail riutha la gu la. agus mur a tiondagh mi gu sligh an fhirin agus mo trocair ann an uine aithghearr bithidh mi air mo ghearradh o bharr an Talbhain agus air mo thilgeall ann an slochd a leir sgrios.

> [I am growing very tired of this place, among wicked people without fear of God or demons, no care for eternity or good in their souls any more than their beasts in the field, and I am sorry to say that I myself am becoming more similar to them day by day. And unless I turn to the path of truth and mercy soon, I will be cut from the Earth and thrown into a pit of destruction.]

This lament may as well have been a premonition; by the time of the second diary, he writes *about* Gaelic more than *in* it. The language does reappear, but only as short phrases such as "Beurla an donais" rather than the detailed, page-long entries of the first diary, and only under conditions of considerable emotional distress. His newfound position as Pākehā, then, never fully subsumed his own Gaelic identity.

However, it is readily apparent that Neil failed to transmit Gaelic language and/or identity to his children. On 5 October 1886 he criticises (in English) his children's public school for leaving their "ears poisoned and young minds almost corrupted", possibly in part because it did not feature their heritage language at all[4]. Given that neither Rebecca nor Neil's second wife Elizabeth appear to have had any Gaelic themselves to pass on following Neil's death in 1890, their children's schooling played an additional role in our family's Gaelic language and therefore identity loss, consolidating subsequent generations' assimilation into the Pākehā world.

### 4.2. Beatha Làitheil/Daily Life in the Armed Constabulary

Neil spent his initial years in Aotearoa mining gold in Pārāwai/Thames with his family. He would eventually leave them behind in 1871, however, to join the New Zealand Armed Constabulary (Smyth 1965, p. 8), a policing, detective, and military force established in 1867 to shore up the colony's enforcement capacities following the departure of British troops sent to fight hapū and iwi Māori[5] in the New Zealand Wars (Hill 1989). Neil dedicates much of his writing in the first diary to the everyday tasks involved in being a Constable at the redoubt in "Alexandra" (the colonial name at the time for Pirongia), where he spends most of the 1870s. In this role, he initially describes consistently having to cook for his colleagues (to the point that on 28 February 1873, he simply writes "I am giving up cooking as I am quite tired of it" and never mentions it again), as well as being part of "Road Parties" comprised of other Constables responsible for building the road infrastructure in the region. Regarding the more enforcement-oriented tasks involved with the role, he frequently details the arrests of vagrants and thieves, sometimes having to involve other Constables in chasing them down, as well as escorting them to court in Hamilton and/or Auckland and occasionally taking them to the gaol as well.

Much of this picture of daily life in the AC is, of course, consistent with its stated purpose to "keep the peace" (Neil's words), but is also revealing of its *inferred* purpose: to "impose the norms of the conqueror upon the conquered" (Hill 2000, p. 34). While building the infrastructure of the colonial state certainly contributes to this imposition, what is even more stark is the way Neil describes Māori individuals who run afoul of the

imposed Pākehā law. Though he does arrest numerous Pākehā as well, it is his descriptions of Māori that are especially revealing of how he positions himself in the settler-colonial context. Quite bluntly, while he does often refer to Māori as Maori (sans macron), equally as often he refers to them by the n-word:

> 8th Saturday [March 1873] It is rumoured that his Excellency the Governor is coming here to pay a farewell visit to the n*****s[6].

> 2nd [May 1873] Dismissed at 6 a.m. dressed and after having breakfast went to see the settlement. saw nothing very striking but heard any amount of rumours about n*****s and daily expected to have a march through the Maori country.

> 21st [June 1876] John Ferugson from Whata Whata laid an information and obtained a warrant for the arrest of a Maori and a halfcaste for stealing. the Half caste was arrested, but the n***** got away, leaving a long start of James Boon who gave him chase so far as the Brick Kiln.

> January 22nd [1878] Const. Gillies came here with a N***** from K.K. who got one month to Mt. Eden and I hope I shall have the pleasure of going with him.

This word choice signals the deployment of an anti-Black racial frame, evoking a set of stereotypes, prejudices, ideologies, narratives, and imagery designed to racialise and marginalise Black peoples (Feagin 2020). To be clear, its use here does not necessarily mean Māori were considered Black at the time; ultimately, they would not become positioned as such within the racial hierarchies imposed on the Pacific from the nineteenth century onwards (Norris and Mire 2022, pp. 199–200; Rew 2022). However, given that anti-Blackness comprises the oldest and consequently most developed racial frame, it has become the primary frame of reference used to produce racialised Others on a global scale (Feagin 2020, pp. 67–68; Norris and Mire 2022, pp. 199–200; O'Malley 2012, p. 230). Neil's use of the n-word to describe Māori therefore demonstrates how he uses his existing familiarity with anti-Black racial framing to make sense of the marginal social position of Māori in this new settler-colonial context, which is consistent with the inferred purpose of the AC to dominate Māori.

It is important to note here that in the nineteenth century Gaels were themselves still racialised Others to an extent, but unlike Pacific peoples were very much in the process of becoming white. This was eventually achieved through participation in imperial projects, such as those in Aotearoa, where colonisation produced opportunities to relocate themselves within a new racial hierarchy (Ignatiev [1995] 2012; MacKinnon 2008; Newton 2008; Newton 2015, p. 35; Stroh 2016, p. 16). What Neil's diaries offer, then, is a demonstration of some of the ways not-quite-white peoples, such as Gaels, made use of the nascent racial structures being established in the Pacific to distance themselves from other colonised and racialised peoples.

If the diminished roles of Gaelic language, culture, and people in Neil's life were amongst the earliest factors contributing to his journey to becoming Pākehā instead, then this deployment of racist tropes indicates what they were gradually replaced by, and for what purpose.

### 4.3. Na Sgeulachdan "Mòra"/the "Big" Stories: Pākehā-Gaels and the Wider World

Periodically throughout his diaries, Neil also writes about national and global affairs. While he comments frequently about domestic proceedings such as the activities of George Grey[7], he appears equally interested in international events such as the Russo-Turkish war, the Prussian-Austrian rivalry, and, most relevantly to the research at hand, the Fenian movement for Irish independence. The Fenian movement took form both domestically and throughout the Irish diaspora, including in Aotearoa where the historical record is defined by largely unsubstantiated "reports" of Māori–Irish collaboration against the colonial state (Bender 2017). Neil, it would appear, was potentially privy to the details of these reports from a firsthand source:

> [12th October 1877] Michael O'Connor the great Fenian came here [to Alexandra] about the 1st Sept last and went to the King Country. it is reported that he is to place some Fenians at the disposal of the n*****s all ready armed to fight against the [host?].

> 17th [October 1877] O'Connor has been up the King Country for the past few days and came down this evening, I had a talk to him, he states that all arrangements are nearly completed for gold digging.

> November 12th [1877] . . . Fenian O'Connor has been paying his respects to Te Naku and Manuhiri, the rascal is trying to form a Fenian colony among the Natives, he told me more nonsense than any rational being I ever spoke to.

Neil's dismissive attitude toward "Michael O'Connor the great Fenian" and his "nonsense" highlights not only an apparent ideological disagreement over the morality of the Empire, but is also consistent with the "big" history of how the Scottish and Irish experiences of subjugation by the British Empire diverged. The Plantation of Ulster in the seventeenth century, through which Ireland was first colonised by the British, intentionally split the Gaelic world in two, cutting Gaels in the Scottish Highlands and Islands off from their countryfolk down south (Stroh 2016, p. 40). From here, for a variety of reasons outside the scope of this article, the Empire began treating Scottish and Irish Gaels differently (Stroh 2016, pp. 38–39); while comparatively Irish Gaels were subjected to more overtly violent and denigrative colonising practices, Scottish Gaels became earlier targets for assimilation (Stroh 2016, pp. 58–59, 62). As a result, by the nineteenth century Scottish Gaelic cultural expression became more tolerated as it had already been largely appropriated by, assimilated into, and identifiable with the Empire; in contrast, Irish cultural expression became more closely linked to a radical nationalism that in many cases sought to challenge that Empire (Stroh 2016, p. 128; McMahon et al. 2017). While it is well-documented that the New Zealand Armed Constabulary was modelled after its Irish predecessor (Hill 2010), and many Irish Gaels also assimilated into Empire and served in the AC like Neil (Shaw 2021a, 2021b), Irish history in the colonial era nevertheless includes more examples of overt resistance to Empire than Scottish Gaelic history[8], especially in the diaspora (McMahon et al. 2017; Stroh 2016).

The results of these divergent histories certainly *correlate* with Neil's description of his interactions with O'Connor. However, they only suggest his attitude toward the Fenian movement and its politics, and do not necessarily paint a sufficiently comprehensive picture of just how distanced Neil had become from his Irish relations more generally. The only other passage in his diaries in which he alludes to the Irish at all is therefore illustrative:

> [13th March 1877] caileag sgiobalta, tha i gle [x] agus riamhar, mu thiomchall coig troighean agus deich iorlaich a dh' airde, agus gle boidhach. tha barail agam gur a than do dh' Eirin a bhuinis i, ach tha mi an dochas nach buin i do dh Eaglais na 'n Roimh . . . ged tha mi cinnteach gun gabh mi mias mor orra ann an uine aithghearr cia dhen a bhuinis i do dh Eaglais Dhaor na Saor.

> [A clever girl, she is very [x] and fat, about five feet and ten inches tall, and very beautiful. I believe she came from Ireland, but I am hopeful that she does not belong to the Catholic Church . . . though I am sure that I will soon have great admiration/affection for her, whether she belongs to the Free Church.]

In this description of his first impressions of his future wife, Rebecca, Neil expresses disapproval towards Catholicism. The development of a British identity in the eighteenth and nineteenth centuries laid "special stress on common enemies", which included the Catholic Church and by extension Gaeldom since it had historically been primarily Catholic (Newton 2019, p. 39). Consequently, missions led by the Presbyterian Church/Free Church of Scotland and the aforementioned SSPCK converted many Scottish Gaels, including Neil and/or our shared ancestors, to a more British faith, further exacerbating the historical divide between Irish and Scottish Gaels by reshaping the latter's cosmology (MacInnes 2006, pp. 439–40; Newton 2019, p. 40; Stroh 2016, p. 66). Neil's remarks in the above

passage reflect this history by demonstrating the extent to which his affiliation to the British Empire vis-à-vis Presbyterianism/the Free Church contributes to his rejection of any (non-romantic) ties to Irish Gaeldom.

I have now demonstrated how Neil struggles to reconcile his identity as a Gael with his new situation in the settler-colony; how this struggle is accented by his disavowal of any affinity toward Māori as colonised peoples through his use of anti-Black racial framing; and how he even disavows any affinity toward his Irish cousins, ignoring the common cause against imperialism in both Irish and Māori cases despite O'Connor allegedly spelling them out for him. In the following discussion, I will now consider how these findings make sense of how Neil "settled" into Aotearoa New Zealand by actively participating in colonial dispossession, despite being dispossessed of his own linguistic, cultural, and terrestrial heritage.

## 5. Discussion

While the many ethnicities comprising the British Empire would eventually become subsumed into whiteness in the settler-colonial context (Bell 1996; Dean 2018; Ignatiev [1995] 2012; Pearson 1989), Neil's dual identity as Gael and Pākehā demonstrate how this was not always the case. Whether it was as a footsoldier in the AC like him, or a more white-collar role, such as that of the more well-known New Zealand land claims commissioner Donald MacLean (Ward 1990), Gaels occupied a range of positions within the Empire—and participated in the dispossession of other peoples from those positions—while still clinging to some form of distinct ethnic rather than just national identity (Newton 2019, p. 39). In this sense, then, it is clear that in the earlier generations of colonisation, non-Anglo ethnic identities within the Empire were not seen as contradictory to participation in it (MacKenzie 2008), and were even accommodated to some extent as the Scottish Gaelic case shows—at least, in a reconstructed form as the model civilised-savage, as discussed above. Correspondingly, there is nothing in Neil's diaries which identifies any overt instance of discrimination as a Gael in Aotearoa New Zealand; at best, his observations about his assimilation and the context of the Highland Clearances which precipitated his emigration can only infer such experiences.

What does not need to be inferred, however, is how Neil transitioned from colonised to coloniser. As his diaries so effectively demonstrate, his displacement from his ancestral homeland and repositioning in the settler-colonial context had a gradual but noticeably destructive effect on the centrality of Gaelic to his identity, reduced as it becomes the language of only his innermost thoughts. In addition to being lost as a community language upon emigration, much of the place-based and culturally-specific knowledge embedded within Gaelic language and culture as Neil experienced it would have had little purpose outside of the *Gàidhealtachd* (Berkes 1993; Chiblow and Meighan 2022; Newton 2015); the resulting "gaps" in Neil's knowledge as they applied to Aotearoa New Zealand would then have been filled by English ways of knowing and being. These would largely have been transmitted through the English language, which by its very structure had already become distinctly colonial and imperial by the nineteenth century (Meighan 2020, p. 2; Smith 2012; Stewart 2020, p. 23). Accordingly, Neil's behaviour as he transitions into a more anglophone life includes the deployment of racist framing against Māori and the rebuke of (Irish) Gaelic interests which explicitly contradict those of the British Empire. There is a strong correlation, then, between the shift in the language he uses in daily life, and his transformation from a Gael of Raasay into a Pākehā settler. Although there is no obvious point at which one becomes the other within the narrative of the diaries, given the evidence it seems more likely to be Neil as-Pākehā who had no observable qualms doing to Māori what had been done to him and his own people.

The long decline of the Gaelic world through experiences such as Neil's, which began before but were exacerbated by the Highland Clearances, could indeed be argued to constitute a historical trauma that Gaels have been subjected to over generations (Mohatt et al. 2014). The Clearances decimated Gaelic culture, scattered most of the people

across the world, and left behind little more than the symbology of tartans, bagpipes, and Highland Games, emptied of meaning and repurposed by the British Empire (Newton 2015, pp. 68–120; Newton 2019, p. 40). As Borell et al. (2018) demonstrate, however, such traumas need to be measured, especially in the settler-colonial context of Aotearoa New Zealand, against Pākehā-Gaels' material *gains* here, and how we have accumulated those gains over generations at the expense of the Indigenous people of this place. Such gains, and their transmission through inheritance, play a determining role in descendants' greater socioeconomic status comparative to Māori (Borell et al. 2018, p. 28). While the process by which this material wealth is transmitted to subsequent generations is more thoroughly demonstrated elsewhere (Bell 2020; Shaw 2021a; Shaw 2021b; Sleeter 2014), it is useful to briefly point out here what Neil gained by becoming Pākehā, not just what he lost. Namely, he acquired significant footholds through the purchase of land in Mahurangi/Warkworth, and later a Government Life Insurance policy in 1885, wealth which in both cases helped cushion the family upon his sudden death by ensuring they were not left destitute by the loss of their breadwinner. Ultimately, that wealth was accumulated through his AC wages (the "little" story) and through the larger role the AC played in extracting that wealth from Māori people and land (the "big" story of colonisation), and over time played a role in ensuring greater financial security for subsequent generations of our family.

Measuring historical privilege against historical trauma like this is a fraught task, however. While inextricably linked (Borell et al. 2018, p. 27), historical privilege is primarily concerned with measuring material wealth; historical trauma can be measured by material deprivation, but is equally capable of considering other outcomes of violent and/or disruptive experiences, such as the loss of ethnic identity, culture, and/or language (Gone 2013, p. 687; Prussing 2014). This mismatch is on full display in Neil's story; though through it I might lay claim to the loss of Gaelic language, culture, and identity, and argue that the circumstances of my ancestors' emigration from Raasay indicate historical trauma, at the same time I must reckon with how Neil's participation in the systematic theft of land and resources from hapū and iwi Māori has historically privileged myself and his other descendants. So long as being colonised is characterised by this material dispossession (Coulthard 2014, pp. 13–14), it is misleading if not outright dishonest as a settler to evoke any historical trauma, any harm *received*, without acknowledging historical privilege in the same breath, and the harm *caused* to attain it. In other words, it is simply not possible to *be* both coloniser and colonised at the same time (Bell 1996, p. 156), even with a family history that contains both.

## 6. Conclusions

It is through critical family history research methods that I have woven together this understanding of Neil McLeod's transformation from Gael to Pākehā. The process by which I have retraced these steps has had to be more thorough than this article can encompass, not only for purpose of academic rigour but also because so much of this knowledge was lost to successive generations. While my family had some vague awareness that we came from the typical mixture of English/Scottish/Irish peoples who comprised the original Pākehā migration waves of the nineteenth century, we had long lost any sense of distinctly Gaelic heritage or identity—not to mention the language. As Borell et al. (2018), Bell (2020), and numerous other scholars have pointed out, this "constitutive forgetting" of how we *became* Pākehā consolidates the Pākehā identity, and all the settler-colonial baggage it entails. As Bell argues, it is imperative that we resist this forgetting which causes us to usurp the Indigenous claims of Māori through an ignorance of our own origins and the violent history of how we came to be here (Bell 1996, p. 156).

Resisting this forgetting has introduced a new challenge, however. While I have sought to be upfront throughout this article about Neil's role in the colonisation of Aotearoa, I also cannot ignore the profound losses that he and our shared forebears experienced through the prior colonisation of the *Gàidhealtachd* via the Highland Clearances, and how that bereavement played a direct role in repositioning us as colonisers. Making meaning out of

*that* legacy, of both harm caused *and* received, appears an exercise in contradiction; while my everyday life today may be characterised by alienation from my Gaelic roots on the one hand, it is equally defined by relative material comfort at the expense of Māori on the other, and has been for six generations. Finding the most ethical way forward while holding both these truths is no small task.

Centuries separate the beginning of the colonial project in the *Gàidhealtachd* from that in Aotearoa or even in Ulster, creating divergent histories which, in Neil's case, contributed to his inability to see common cause with his Māori or Irish neighbours. What critical family history research offers us today is the advantage of greater hindsight: our collective experiences of colonisation today, on every side of it, may be incomparable, but the historical forces which repositioned us on opposing sides came from the same place. Acknowledging that must surely be the basis for a shared understanding through which we might find a more stable common ground.

**Funding:** This research received no external funding.

**Data Availability Statement:** The primary data for this article come from the personal diaries of the author's ancestor, Neil McLeod. Physical copies of the diaries are publicly available in the library archives at the Auckland War Memorial Museum as the singular item MS-182.

**Acknowledgments:** The author would like to thank the peer reviewers, for their insightful feedback on earlier drafts of the article; the editors Giselle Byrnes and Catharine Coleburne, for their attentive guidance and curation of this special issue; Richard Shaw, for the encouraging conversations that helped inspire the final analysis; agus an sinn-seachad-sinn-seanair, Niall MacLeòid, airson cho mòran a sgrìobhadh mu dheidhinn a bheatha. Gabhaibh fois gu math, a sheanair.

**Conflicts of Interest:** The author declares no conflict of interest.

## Notes

[1] Some of the pages contained in the photocopy I was able to access were too faded to read for transcription and/or translation.

[2] Colonialism and capitalism were far from the only contributing factor the Clearances; changes at the time in agricultural practice and improvements in medicine caused an enormous population boom across Europe. (Newton 2019, p. 37).

[3] While in modern Gaelic the word Beurla refers specifically to English, in older forms of the language (possibly including Neil's) it is a closer analogue to the word 'language' itself (MacBain 1982).

[4] The only school known to have ever taught in Gaelic in Aotearoa is in Whangārei Heads, though it only did so for a few years up until roughly around the time Neil first arrived in Aotearoa (Howell 2007, pp. 26–30).

[5] Māori subtribes and tribes.

[6] Given contemporary sensibilities, and with respect to mine and Neil's non-Black positionality, I have opted to censor the n-word for the purposes of this article. To be clear, it is not censored in the source text.

[7] Both as Governor and not; the diaries cover a long enough span to touch on multiple periods of Grey's career.

[8] Which is not to say that Scottish Gaels never resisted; for example, the Highland Land League of the late nineteenth century fought for and eventually achieved legislative changes through the Crofters Holdings (Scotland) Act 1886 that would, in the legal arena at least, formally end the Highland Clearances (Newton 2019, pp. 41–42; Thomson 1983).

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
