# Peer review of "Harm Received, Harm Caused: A Scottish Gael’s Journey to Becoming Pākehā"

_genealogy, doi:10.3390/genealogy6040082_

Round 1

Reviewer 1 Report

The paper is a good one but could be improved as follows:

1. The paper does not have a strong explanation of the distinction between class and ethnicity.  The historical description is a bit brief and does not explicate the historical change from small farm to large landholdings and who was involved with that--Who were the landholders in the highlands and their relationship to the Anglo-Saxon.  How was it any different than in England itself. Where and when does it crossover to "othering," and who does the othering?

2.  Using the journal entries is really good, but at times the "results" seem over-generalized. The paper could use some refining of its main points with a focus on those most important to the main claim with backing from sources plus a wider number of quotes that support the claim.

Author Response

Firstly, I want to thank reviewer 1 deeply for their input. In hindsight, of course, it is abundantly clear that my ‘highly condensed overview of Scottish Gaelic history’ was in fact too condensed in its elision of who was othered, who was responsible, how class and ethnicity interplay in this context, and how agricultural/landholding developments factoring into this history are significant enough to warrant inclusion. Revisions to this section incorporate clearer explanations of all the above (as well as further ethnicity/class distinctions), particularly between lines 189-218.

Regarding feedback pertaining to the results, I reassessed how some of the claims were made and could see what reviewer 1 meant in terms of claims being too over-generalised/broad. For example, claims between lines 430-451 now avoid claims about how anti-Black racial framing was imported to Aotearoa wholesale, but specifically how *Neil* used that framing and how that still contributes to the central claim through a narrower and more practicable scope. As per feedback, more excerpts from the diary have been included in all three subsections, and more sources have been included to support arguments too, particularly around anti-Blackness in the Pacific, the intertwining of language with worldview and identity, which appeared comparatively underdeveloped.

Reviewer 2 Report

I enjoyed reading this article.  My only suggestion would be to include something more on Neil's literacy in both English and Gaelic. ( a few sentences?) His literacy and bi-lingual ability is a central element of your argument. Is there any information that can be gained about schooling in this remote area of the Highlands in the mid 19th century?  The Scots valued education, so perhaps some comment there would assist if there is no specific information on the education system where he grew up. I think his childhood predates compulsory schooling?  He is in his mid-twenties when he arrived in NZ so as an adult had probably used his bi-lingual literacy prior to emigration.

Author Response

Firstly, I want to thank reviewer 2 for this encouraging feedback. I did indeed have more information about the nature of Scottish education in this part of the Highlands at the time Neil was there, and have added an additional paragraph between lines 242-252 to flesh out that aspect of Neil’s background in order to better set up his journaling. In short, he did receive formal education, and it was a product of colonial tension between the anglicising forces driving that education, and the increased Gaelic literacy it ironically necessitated in practice. Evidence specific to Raasay and therefore Neil is also included in these sources.